

# Arrested development and fragmentation in strongly-interacting Floquet systems

**Matthew Wampler$^\star$ and Israel Klich**

Department of Physics, University of Virginia, Charlottesville, Virginia 22903, USA

$\star$ mbw5kk@virginia.edu

## Abstract

We explore how interactions can facilitate classical like dynamics in models with sequentially activated hopping. Specifically, we add local and short range interaction terms to the Hamiltonian and ask for conditions ensuring the evolution acts as a permutation on initial local number Fock states. We show that at certain values of hopping and interactions, determined by a set of Diophantine equations, such evolution can be realized. When only a subset of the Diophantine equations is satisfied the Hilbert space can be fragmented into frozen states, states obeying cellular automata like evolution, and subspaces where evolution mixes Fock states and is associated with eigenstates exhibiting high entanglement entropy and level repulsion.

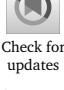

# 1   Introduction

As experimental tools have progressed (e.g. [1,2]), the microscopic control of quantum systems has become increasingly accessible. These advancements, along with a correlated increase in theoretical interest, have led to the discovery of many new and surprising phenomena that emerge when periodic driving, interactions, and their interplay are considered.

For example, periodically driven systems can be used to stabilize otherwise unusual behavior. A recent important example is topological Floquet insulators [3–5], where novel topological features of the band structure may emerge due to inherent periodicity of the non-interacting quasi- energy spectrum. Furthermore, it was shown in [6] that, by combining spatial disorder with a topological Floquet insulator model introduced by Rudner-Lindner-Berg-Levin (RLBL) [7], a new topological phase may be realized called the anomalous Floquet-Anderson insulator (AFAI). Discrete time crystals [8–11] are another important example of behavior that may occur in periodically driven, but not static [12], systems. Namely, a time crystal is a system where time-translation symmetry is spontaneously broken (in analogy to spatial translation symmetry spontaneously breaking to form ordinary crystals).

Combining periodic driving with interactions, however, can often be problematic as generic, clean, interacting Floquet system are expected to indefinitely absorb energy from their drive and thus quickly converge to a featureless infinite temperature state [13–15]. This problem may be side-stepped by considering Many-Body Localization (MBL) [16–22], in which strong disorder is utilized to help stave off thermalization, by considering the effective evolution of pre-thermal states [23–28] that, in the best cases, take exponentially long to thermalize, or by connecting the system to a bath to facilitate cooling and arrive at interesting, non- equilibrium steady-states [29–32].

Yet another route for realizing non-trivial dynamics despite the expected runaway heating from interacting, Floquet drives is to consider systems where the ergodicity is weakly broken, i.e. where there are subspaces (whose size scales only polynomially in the system size) of the Hilbert space that do not thermalize despite the fact that the rest of the Hilbert space does. These non-thermal states are called quantum many-body scars [33–35] and have been shown to support many interesting phenomena including, for example, discrete time crystals [36]. Furthermore, in constrained systems, the full Hilbert space may fragment into subspaces where some of the subspaces thermalize while others do not [35,37,38]. When the fraction of non-thermal states are a set of measure zero in the thermodynamic limit, the system is an example of quantum many-body scarring. However, in other cases, the non-thermal subspaces form a finite fraction of the full Hilbert space and therefore correspond to a distinct form of ergodicity breaking.

In addition to leading to heating, interactions are also often responsible for our inability to efficiently study or describe many body quantum states in both Floquet and static Hamiltonian systems. However, there are situations when interactions play the opposite role in creating specialized states of particular simplicity or utility. For example, systems with interactions can exhibit counter-intuitive bound states due to coherent blocking of evolution. A nice class of such systems are the edge-locked few particle systems studied in [39,40].

In this work, we consider Floquet drives where hopping between neighboring pairs of sites are sequentially activated. The theoretical and experimental tractability of such models have made them a popular workhorse for fleshing out a broad range of the exciting properties of periodically driven systems (e.g. [7, 41–45]). We find that, when interactions are added to such systems, there exist special values of interaction strength and driving frequency where the dynamics becomes exactly solvable. Furthermore, the complete set of these special parameter values may be determined via emergent Diophantine equations [46]. At other parameter values, the Hilbert space is fragmented. Initial states contained within some, thermal, subspaces will ergodically explore the subspace (though not the entire Hilbert space), while other initial states contained within other, non-thermal, subspaces will evolve according to a classical cellular automation (CA) [47,48], i.e. the system evolves in discrete time steps where after each step the occupancy of any given site is updated deterministically based on a small set of rules determined by the occupancy of neighboring sites.

As examples, we consider RLBL(-like) models with added nearest neighbor (NN) or Hubbard interactions as well as an even-odd Floquet drive in one dimension with NN interactions (more detailed descriptions of these models given below). We note that some work has been done in the first two cases [49,50] where it was argued that novel, MBL anomalous Floquet insulating phases emmerged when a disorder potential was added. We will discuss how our focus on special parameter values leads to new insights into these models and how it suggests a possible route towards other exciting phenomena such as the support of discrete time crystals within fragments of the Hilbert space.

## 2 Conditions for evolution by Fock state permutations

In this section, we examine conditions for deterministic evolution of Fock states into Fock states in fermion models. Here we consider real space Fock states, which have a well defined fermion occupation on each lattice site (We will also refer to such states as fermion product states). We consider models where hopping between non-overlapping selected pairs of sites is sequentially activated. Two models of this type, discussed in detail below, deal with Hubbard and nearest neighbour interactions. The approach can be naturally extended to deal with more general interactions in sequentially applied evolution models.

### 2.1 Example 1: Hubbard-RLBL

As a particularly illuminating example, consider the Rudner-Lindner-Berg-Levin model [7]. This model is an exact toy model for a topological Floquet insulator and has been very useful in flushing out some of their salient properties. In addition, it provides the starting point for other states, such as the anomalous Floquet-Anderson insulators [6]. The model is two dimensional, however, it's simplicity lies in its similarity to even-odd type models, [41,43–45], in that the evolution activates disjoint pairs of sites at each stage. The model can be tuned to a particular point where the stroboscopic evolution of product states is deterministic exhibiting bulk periodic motion and edge propagation. Similarly, one can tune the driving frequency to completely freeze the stroboscopic evolution. Here, we add interactions to the model and ask when we can make the evolution a product state permutation, at least in some sectors. The Hubbard-RLBL evolution is written as

$$U = U_{wait}U_4 U_3 U_2 U_1 \,, \tag{1}$$

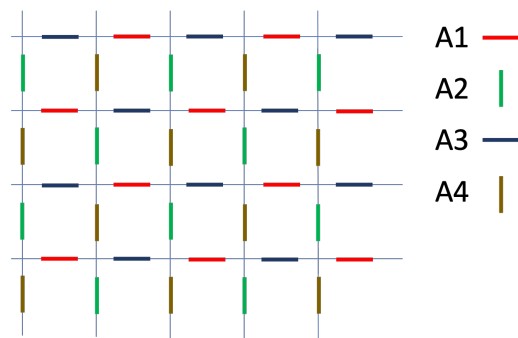

Figure 1: The RLBL model. Hopping is sequentially activated among neighbouring sites connected in the set $A_i$, $i = 1, ..., 4$.

where $U_i(V, \tau) = e^{-i\tau H_i}$. For $i = 1, ..4$,

$$\mathcal{H}_i = -t_{hop} \sum_{(i,j) \in A_i; \sigma} (a_{i,\sigma}^\dagger a_{j,\sigma} + h.c.) + V \sum_{i \in A_i} n_{i,\uparrow} n_{i,\downarrow}, \tag{2}$$

where $n_{i,\sigma} = a_{i,\sigma}^\dagger a_{i,\sigma}$ and the sets $A_i$ are described in Fig. 1. Throughout the rest of the paper we will work in units where $t_{hop} = 1$ and $\hbar = 1$.

Above, $U_{wait}$ is any unitary diagonal in number state basis. For example, the model investigated in [50] has $U_{wait} \rightarrow U_{dis}$ where $U_{dis}$ corresponds to evolution under the Hamiltonian $\mathcal{H}_{dis} = \sum_i v_i(n_{i,\uparrow} + n_{i,\downarrow})$ with $v_i$ a vector of uniformly distributed random real numbers within the bounded interval $[-W, W]$, i.e. the waiting period corresponds to evolution with a disordered on-site potential and no hopping.[1] In that work, it was shown that this model supports a new family of few-body topological phases characterized by a hierarchy of topological invariants. These results may be viewed from the following perspective. First, finely-tuned points where the dynamics is exactly solvable were studied (namely, $\tau = \frac{\pi}{2}$ and $V = 0$ or $V \rightarrow \infty$). Second, it is argued that regions near these special points are stabilized (i.e. localized, at least for finite particle number cases) by disorder leading to robust phases. Finally, topological invariants characterizing these phases (V small vs. V large) can be found and shown to be distinct implying two differing topological phases. An application of the methods we propose in this work will allow us to generalize the first step above and find families of these exactly solvable points. We leave discussions of when regions in parameter space near these points may or may not be stabilized by disorder to future work. Since, at these exactly solvable points, we will be mapping product states to product states, $U_{dis}$ will only act as an unobservable global phase and thus for the rest of our analysis we will set $U_{wait} = I$.

We now look for conditions to simplify the evolution (1) in such a way that the total evolution reduces to a permutation on the set of product states, i.e. when an initial configuration of fermions is placed at a selection of locations it will evolve into a different assignment of locations without generating entanglement.

To do so, we note that the evolution of each pair of sites, may be considered separately due to the disjoint nature of the set of pairs $A_i$. Thus, we consider the evolution on a pair of sites $i, j$

$$U_{(i,j)}(V, \tau) = e^{-i\tau(a_{i,\sigma}^\dagger a_{j,\sigma} + h.c.) + \tau V(n_{i,\uparrow} n_{i,\downarrow} + n_{j,\uparrow} n_{j,\downarrow})}. \tag{3}$$

---

[1]Technically, in [50] a weak disorder potential is added during the $U_i$ steps and then the disorder strength during the wait step is effectively made stronger by increasing the length of time the wait step is applied. However, this slight difference in how the disorder potential is applied does not seriously alter the dynamics and so we will not make a hard distinction between the two.

Since the evolution preserves particle number, we can treat the sub-spaces of $0, 1, 2, 3,$ and $4$ particles in each neighboring pair of sites separately. In the case of 0 or 4 particles, evolution is trivially the identity (due to Pauli blocking in the 4 particle case). For 1 or 3 particles, one of the two sites is always doubly occupied, and thus the interaction term in (2) is a constant and does not affect evolution. In this case, solving the two site non-interacting evolution we see that in the one-particle sector, a fermion starting initially at site $i$ has a probability $p = \sin^2 \tau$ to hop to the other site in pair $j$ and probability $1-p$ to stay. Similarly, in the 3-particle sector, an initially placed hole in site $i$ has the same probability, $p$ to hop to the other site $j$. Thus, when

$$\tau = \frac{\pi}{2}\ell\,, \tag{4}$$

for some integer $\ell$, evolution for initial product states in the 1,3 particle subspace is completely deterministic with trivial evolution for even $\ell$ and the particle hopping to the other site in the pair with probability 1 (henceforth referred to as perfect swapping) when $\ell$ is odd. Clearly, for these values of $\tau$ (and independently of $V$), no new entanglement is created in any pairs with 1 or 3 particles. To render the evolution in the 2 particle pair subspace simple, it is shown in appendix A.1 that deterministic evolution occurs when the two conditions below are simultaneously satisfied:

$$\tau\sqrt{4^2 + V^2} = 2\pi m\,, \tag{5}$$

and

$$\frac{1}{2}\tau V + \pi m = \pi n\,, \tag{6}$$

with $n, m \in \mathbb{Z}$. Note that (5) guarantees the preservation of the number of doubly occupied sites (doublons). When $n$ is even, the sub-system will return to its initial state. On the other hand, if $n$ is odd, the system will exhibit perfect swapping i.e. each particle will hop to the other site in the pair. By solving for $\tau$ and $V$ in terms of $n$ and $m$, we may now summarize when evolution is deterministic in each of the particle number sub-spaces:

| particles | $\tau$ | $V$ |
|---|---|---|
| 1 or 3 | $\tau = \frac{\pi}{2}\ell$ | $V$ arbitrary |
| 2, opposite spins | $\tau = \frac{\pi}{2}\sqrt{2mn - n^2}$ | $V = \frac{4(n-m)}{\sqrt{2mn-n^2}}$ |
| otherwise | any | any |

(7)

when $n$ or $\ell$ are even (odd) evolution is frozen (perfect swapping). To keep the solutions real, Eq. 7 also implies we must take $2mn - n^2 > 0$.

Can all the conditions (4), (5), and (6) be simultaneously satisfied? In such a case the evolution of $\mathcal{U}$ is simply a permutation (being a product of identities and site swaps) and generates no new entanglement in any of the sectors.

## 2.2 The Diophantine Equation

Combining the conditions (4), (5), and (6) together yields the following equation:

$$\ell^2 + n^2 = 2mn\,, \quad \ell, n, m \in \mathbb{Z}\,. \tag{8}$$

Eq. (8) is a homogeneous Diophantine equation of degree 2 and can be solved.

We now give a brief review of Diophantine equations and the strategy for solving homogeneous quadratic equations. A reader familiar with Diophantine equations or interested only in the concrete results may skip to the next subsection.

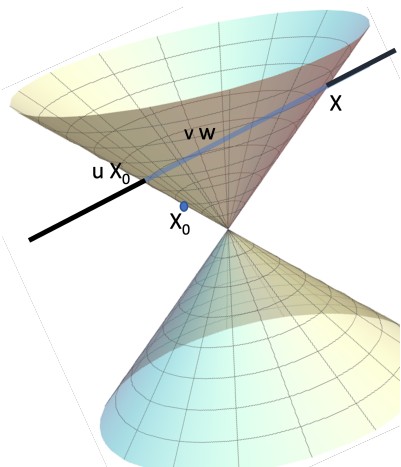

Figure 2: Any line passing through the null surface has two points of intersection. Given a particular solution $X_0$ of the homogeneous Diophantine eq (10), other rational solutions are found by looking at lines emanating from $uX_0$ with rational slopes.

Diophantine equations are algebraic (often polynomial) equations of several unknowns where only integer or rational solutions are of interest. They are named in honor of Diophantus of Alexandria for his famous treatise on the subject written in the 3rd century though the origins of Diophantine equations can be found across ancient Babylonian, Egyptian, Chinese, and Greek texts [46]. Despite their often innocuous appearance, they are an active area of research with solutions frequently requiring surprisingly sophisticated mathematical techniques and have been the centerpiece of several famous, long-standing mathematical problems that have only been (relatively) recently resolved, including Fermat's Last Theorem [51] and Hilbert's Tenth Problem [52].

In this section, we are interested in the relatively simple case of a homogeneous quadratic Diophantine equation, i.e. equation of the form

$$X^T Q X = 0, \tag{9}$$

with variables $X^T = (x_0, x_1, ..., x_n)$ and coefficients given by the $n \times n$ symmetric matrix $Q$ with integral diagonal entries and half integral off-diagonal entries. As we shall see, however, for interactions beyond Hubbard a broader class of Diophantine equations may need to be considered. For information on broader classes of Diophantine equations and for more information on the derivation to follow, see, for example, [46].

The general strategy for finding rational (we will specialize to integer solutions for our cases of interest at the end) solutions to (9) is to first find a particular solution and then generate all other rational solutions from the particular solution. Particular solutions can be found simply by inspection or through existing efficient algorithms [46]. The main task is then to generate all other rational solutions from a given particular solution.

Take $X_0^T = (x_{0,0}, x_{1,0}, ..., x_{n,0})$ to be a particular solution, i.e.,

$$X_0^T Q X_0 = 0. \tag{10}$$

Since (9) is quadratic, any line through $X_0$ will intersect the hypersurface defined by (9) at a single other point (see Fig. 2). Furthermore, if the line through $X_0$ is rational (i.e. has rational coefficients), as we see below, this implies that the second intersection point must also be rational. Therefore, it is possible to generate every rational solution to (9) by finding the second intersection point of every rational line through $uX_0$, where $u$ is rational.

Here, since (9) is homogeneous, it is convenient to work in projective space $\mathbb{P}_n(\mathbb{Q})$ where a general line passing through $X_0$ is parameterized by

$$X = uX_0 + vW, \tag{11}$$

with $(u,v) \in \mathbb{P}_2(\mathbb{Q})$ and any $W = (w_1, .., w_n) \in \mathbb{P}_n(\mathbb{Q})$ not equal to $X_0$. Combining (11) and (9),

$$0 = (uX_0 + vW)^T Q(uX_0 + vW) = v\left(2uW^T QX_0 + vW^T QW\right), \tag{12}$$

where we have simplified using (10). We may thus take as the solution $(u,v) = \left(W^T QW, -2W^T QX_0\right)$. Combining with Eq (11) and multiplying by a general $d \in \mathbb{Q}$ to restore full solutions (since we considered $X$ as an element of a projective space), we find

$$X = d\left[(W^T QW)X_0 - 2(W^T QX_0)W\right]. \tag{13}$$

For integer solutions, we need simply to rescale $W \to \frac{W}{\zeta}$ and $d \to d\zeta^2$ where $\zeta = \gcd(w_i)$. After rescaling, the only non-integer information is coming from $d$, so all integer solutions may be found simply by considering $d \in \frac{1}{\xi}\mathbb{Z}$ with $\xi = gcd((W^T QW)X_0 - 2(W^T QX_0)W)$.

For the relevant case of $n = 3$, let us, without loss of generality, diagonalize $Q = \mathrm{diag}(A, B, C)$ and let $W^T = (w_1, w_2, 0)$ where (after rescaling with $\zeta$) $w_1$ and $w_2$ are co-prime integers and the final element of $W$ may be set to 0 due to the required linear independence with $X_0$. Simplifying (13) then becomes

$$\begin{aligned}
X &= d(Aw_1^2 + Bw_2^2)\begin{pmatrix} x_{0,0} \\ x_{1,0} \\ x_{2,0} \end{pmatrix} - 2d(w_1 Ax_{0,0} + w_2 Bx_{1,0})\begin{pmatrix} w_1 \\ w_2 \\ 0 \end{pmatrix} \\
&= d\begin{pmatrix} -(Aw_1^2 - Bw_2^2)x_{0,0} - 2Bw_1 w_2 x_{1,0} \\ (Aw_1^2 - Bw_2^2)x_{1,0} - 2Aw_1 w_2 x_{0,0} \\ (Aw_1^2 + Bw_2^2)x_{2,0} \end{pmatrix}.
\end{aligned} \tag{14}$$

## 2.3 Solution for product state permutation dynamics with Hubbard interaction

Following the previous section, we write our Diophantine eq. (8) in a diagonal form:

$$\ell^2 + n^2 = 2mn \implies \begin{pmatrix} \ell & \tilde{n} & m \end{pmatrix}\begin{pmatrix} 1 & 0 & 0 \\ 0 & 1 & 0 \\ 0 & 0 & \text{-}1 \end{pmatrix}\begin{pmatrix} \ell \\ \tilde{n} \\ m \end{pmatrix} = 0, \tag{15}$$

where we have defined $\tilde{n} \equiv n - m$. Note, this is the famous Diophantine equation for Pythagorean triples.

By inspection, a non-trivial solution is $\ell = -1, \tilde{n} = 0, m = 1$. Utilizing Eq. (14) we find

$$\begin{pmatrix} \ell \\ \tilde{n} \\ m \end{pmatrix} = d\begin{pmatrix} w_1^2 - w_2^2 \\ 2w_1 w_2 \\ w_1^2 + w_2^2 \end{pmatrix} \tag{16}$$

$$\implies \begin{pmatrix} \ell \\ n \\ m \end{pmatrix} = d\begin{pmatrix} w_1^2 - w_2^2 \\ [w_1 + w_2]^2 \\ w_1^2 + w_2^2 \end{pmatrix}. \tag{17}$$

Note, Eq. (16) is the standard solution for Pythagorean triples.

We thus found that the set of $n$, $m$, and $\ell$ simultaneously satisfying the conditions for simple dynamics can be written as:

$$\ell = d(w_1^2 - w_2^2), \tag{18a}$$

$$m = d(w_1^2 + w_2^2), \tag{18b}$$

$$n = d(w_1 + w_2)^2, \tag{18c}$$

where $w_1, w_2 \in \mathbb{Z}$, $w_1, w_2$ are coprime, and $d \in \frac{1}{\xi}\mathbb{Z}$ with $\xi = gcd((w_1^2 - w_2^2), (w_1^2 + w_2^2), (w_1 + w_2)^2)$. Note, in (18), if $\ell$ is even (odd) then so is $n$. This implies that the only way to completely satisfy the conditions in Eq. (7) is if all motion is frozen or all motion (not constrained by Pauli exclusion) becomes perfect swapping.

Inspecting the above solutions, we see that $2mn - n^2 = (w_1^2 - w_2^2)^2$, automatically satisfying the condition $2mn - n^2 > 0$ for $V$ and $\tau$ to be real. Finally our solution is summarized by

$$\tau = \frac{\pi}{2}d(w_1^2 - w_2^2), \qquad V = \frac{8w_1 w_2}{|w_1^2 - w_2^2|}. \tag{19}$$

Note that $V$ doesn't depend on the choice of $d$, and that any choice involving $w_1 = 0$ or $w_2 = 0$ will yield a non-interacting model. As an illustration, consider the following example choices:
1. Taking $w_1 = 1, w_2 = 0, d = 1$ yields $\tau = \frac{\pi}{2}, V = 0$, which is the non-interacting dynamics considered in the original RLBL model, with perfect swapping.
2. Taking $w_1 = 3, w_2 = 1, d = 1$ yields $\tau = 4\pi, V = 3$. Since $\ell$ is even in this case, the dynamics is completely frozen.
3. Taking $w_1 = 3, w_2 = -1, d = 1$ yields $\tau = 4\pi, V = -3$, i.e. frozen dynamics in a model with an attractive Hubbard interaction.

It is important to note that the special values of interaction strength and driving frequency in Eq. (19) hold for any Hubbard-Floquet procedure where hopping between pairs of sites is sequentially activated. This is the case for such systems on any lattice and in any dimension. We also note, that the Diophantine solution is ill suited to describe the singular case of infinite $V$ and finite $\tau$ and therefore this situation must be handled separately. In the limit of large $V$, the interaction strength overpowers the hopping strength and all evolution is frozen in the 2-particle sector. On the other hand, evolution in the 1,3 particle sector is independent of $V$ and therefore may exhibit perfect swapping or freezing. Thus, in this case, it is possible to have one sector (the 2-particle sector) frozen while the other (the 1,3 particle sector) exhibits perfect swapping.

To visually represent the position of our special points we introduce the following function as a qualitative estimate for how far a given evolution $U$ is from being a permutation of basis states:

$$F_{p,q}(U) = -\log \frac{||U||_{p,q}}{dim(U)^{1/q}} = -\frac{1}{q}\log \frac{\sum_{n,m}|U_{n,m}|^p}{dim(U)}, \tag{20}$$

where $dim(U)$ is the dimensionality, and $||U||_{p,q} = (\sum_{n,m}|U_{n,m}|^p)^{1/q}$ is the $p, q$ matrix entry-wise norm. Note that for a complex permutation, each row has a single non zero entry of absolute magnitude 1, therefore, for any $p, q > 0$,

$$F_{p,q}(\text{complex permutation}) = 0.$$

Note that for any unitary matrix $U$, $F_{2,q}(U) = 0$. However, for $p > 2, q > 0$ we have that $F_{p,q}(U) > 0$ whenever $U$ is a unitary that is not a complex permutation. The following extensivity property is straightforward to verify:

$$F_{p,q}(U_1 \otimes U_2) = F_{p,q}(U_1) + F_{p,q}(U_2). \tag{21}$$

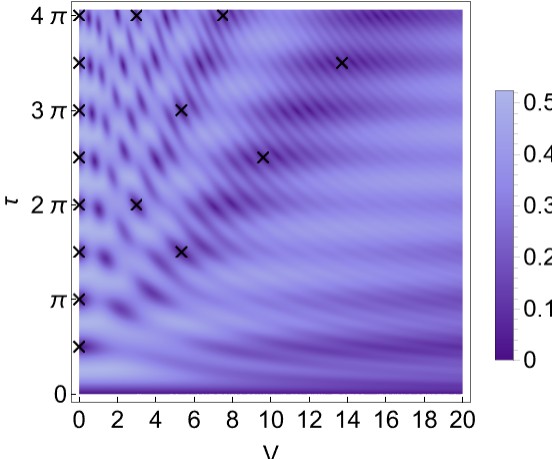

Figure 3: Special Diophantine points in the Hubbard-RLBL model and 'distance' of the two-site evolution from being a permutation of number states. Here we plot $\sqrt{F_{4,4}(U)}$ as function of $V, \tau$. The darker regions indicate regions where the two-site evolution is close to pure number state permutation.

In figure Fig. 3, we plot $F_{4,4}$ for the two site Hubbard evolution showing regions where the two-site evolution is close to permutative and marking the Diophantine spots where it is exact. Interestingly, while the special points admit exact evolution, the plot shows many regions where the evolution is close to perfectly permutative. Exploration of what happens when the evolution is not exact, but perturbatively close to it is out of the scope of the present paper and deserves a separate study, as initiated in [53].

## 2.4 Example 2: Nearest neighbour interactions on a Lieb lattice.

In the next two examples, we consider interactions involving nearest neighbours. Unfortunately, adding nearest neighbour interactions to the RLBL model directly destroys an essential feature for the solvability of the problem: that the evolution operators of different pairs of sites are not directly coupled (and therefore commute). Here, instead, we choose to work with RLBL-like dynamics on a Lieb lattice as described in [54]. A Lieb lattice is a decorated square lattice as shown in Fig. 4. The dynamics we consider here essentially activates pairs that are separated by several lattice sites at each step. The sequence of activations is described in Fig 4.

Here, we consider spinless fermions on the Lieb lattice. There are 8 steps. At step $i$ we activate hopping between sites that are nearest neighbours that belong to the set $A_i$. The evolution is given by:

$$U = U_8 U_7 U_6 U_5 U_4 U_3 U_2 U_1 \,, \tag{22}$$

where $U_i = e^{-i\mathcal{H}_i \tau}$, and

$$\mathcal{H}_i = -t_{hop} \sum_{(i,j)\in A_i} (a_i^\dagger a_j + h.c.) + V \sum_{<i,j>} n_i n_j \,. \tag{23}$$

We proceed, as in Section 2.1, by considering the evolution of a single connected pair during step $i$ and exactly solving for values of $V$ and $\tau$ where the pair exhibits freezing or perfect swapping. The evolution of a 2-site pair of sites $i, j$ for one step is given by

$$U_{(i,j)} = e^{-i\tau[-t_{hop}(a_i^\dagger a_j + h.c.) + V n_i \sum_{k:\langle i,k\rangle} n_k + V n_j \sum_{k:\langle j,k\rangle} n_k]} \,. \tag{24}$$

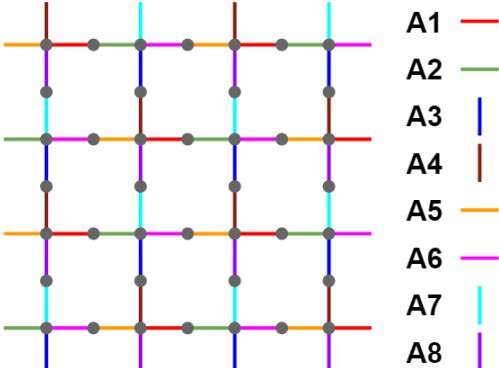

Figure 4: RLBL-like model on a Lieb lattice. Hopping between neighboring pairs of sites within $A_i$ is activated during step $i$ of the Floquet drive. The same sequence of activated site pairs is achieved with the chiral measurement scheme introduced in [54]. During each step $i$, evolution is confined between neighboring sites in $A_i$ by rapidly measuring (in the Zeno limit) all sites in the complimentary set $A_i^c$. Both models, with NN interactions, will share the same conditions (Eqs. (26) and (27)) for number state to number state evolution.

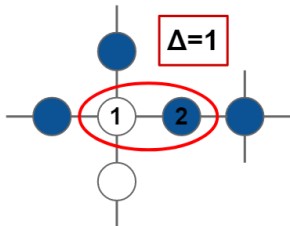

Figure 5: Evolution of a 2-site pair in the NN-RLBL model on a Lieb lattice. All evolution is restricted to the red ellipse above. Evolution within the red ellipse (i.e. between site 1 and site 2) is determined by $\tau$, $V$, and the neighboring particle number difference $\Delta = |N_1 - N_2|$. In this case, $N_1 = 2$ and $N_2 = 1$, so $\Delta = 1$. If the $\Delta = 1$ condition on $V$ and $\tau$ in Eq. (26) is satisfied, then the particle at site 2 will exactly return to site 2 after a time $\tau$ (at intermediate times, the particle may be in a generic superposition of being located at site 1 and site 2).

Note that the number operators on neighbours of $i, j$ commute with the evolution. Let the initial number of occupied neighbours of the sites $i$ and $j$ be $N_i$ and $N_j$ respectively (not counting $i, j$ themselves). Evolution of the 2-site pair is now exactly solvable in terms of $\Delta = N_i - N_j$, the difference in the number of particles neighboring sites $i$ and $j$ in the 2-site pair respectively (see Figure 5).

Solving the two site evolution, we find that evolution is frozen when

$$\sqrt{4 + \Delta^2 V^2}\,\tau = 2\pi m, \tag{25}$$

for some $m \in \mathbb{Z}$. We find that the evolution may only be perfect swapping when $\Delta = N_i - N_j = 0$ or when $V = 0$ and occurs when $\tau = \frac{\pi}{2} + \pi m$ for $m \in \mathbb{Z}$ (see appendix A.2 for details).

In the rest of the paper, whenever considering the evolution on a pair of sites, we will denote $\Delta$ as the difference in the number of (static) particles that are nearest neighbours of the two sites during the relevant evolution step.

## 2.5   A coupled set of Diophantine Equations

For a generic initial position of the particles, $N_i - N_j$ will not be uniform across the sample. Thus, for proper particle permutation dynamics, we must simultaneously find a solution of (25) for all possible values of $|N_i - N_j|$. As we have seen for $V = 0$ there is no dependence on neighbour occupation and evolution will be frozen or perfect swapping if $\tau = \frac{\pi}{2}m$ with m, correspondingly, even or odd. In the rest of the section we concentrate on $V \neq 0$.

Note that $N_i$ takes the values $0, .., D_i - 1$, where $D_i$ is the degree (number of neighbours) of lattice site $i$. It follows that $|N_i - N_j| \in \{0, .., max(D_i, D_j) - 1\}$. Thus, if $D_{max}$ is the maximum degree of the lattice, we have the simultaneous conditions:

$$\sqrt{4 + \Delta^2 V^2}\,\tau = 2\pi m_\Delta \ \forall \ \Delta = 1, ..., (D_{max} - 1), \tag{26}$$

$$\tau = \frac{\pi}{2}m_0 \text{ corresponds to } \Delta = 0 \ (N_i = N_j), \tag{27}$$

with all $m_i \in \mathbb{Z}$.

Equations (26) and (27) provide $D_{max}$ equations that must be solved simultaneously. The first two equations set the values for $\tau$ and $V$ in terms of $m_0, m_1$:

$$\tau = \frac{\pi}{2}m_0, \qquad V^2 = 4\left(\frac{4m_1^2}{m_0^2} - 1\right). \tag{28}$$

However, the rest of the equations for $m_i$, with $i > 1$, must be simultaneously solved with these values for $\tau$ and $V$ yielding the coupled equations:

$$4m_l^2 = (1 - l^2)m_0^2 + 4l^2 m_1^2, \qquad m_l \in \mathbb{Z}, \ l = 2, 3, ..., (D_{max} - 1). \tag{29}$$

A first solution to this system may be obtained by taking $m_0 = 2m_1 = 2m_2 = ... = 2m_{D_{max}-1}$, which, by (28), turns out to be the same as the non-interacting frozen case $V = 0$. We now search for other solutions, with $V \neq 0$.

*Solution for $D_{max} = 3$.* For $D_{max} = 3$, we describe a general solution in appendix A.2 that yields non-trivial solutions. The result:

$$\begin{pmatrix} m_0 \\ m_1 \\ m_2 \end{pmatrix} = d \begin{pmatrix} -32w_1 w_2 \\ -3w_1^2 - 16w_2^2 \\ 2\left[-3w_1^2 + 16w_2^2\right] \end{pmatrix}. \tag{30}$$

We note that $m_0$ resulting from (30) is always even (see the end of Appendix A.2) and thus can only yield frozen evolution when $V \neq 0$. Due to the hierarchy of the equations, total freezing must then occur for any solutions with $D_{max} \geq 3$.

*Solution for $D_{max} = 4$.* We combine equations (30) and the $\Delta = 3$ equation from (26) to find a new Diophantine equation for the case $D_{max} = 4$:

$$\frac{1}{d^2}m_3^2 = 81w_1^4 + 2304w_2^4 - 1184w_1^2 w_2^2. \tag{31}$$

The Diophantine equation (31) is harder to solve. However, a numerical search does find non-trivial ($V \neq 0$) solutions. For example, $(w_1; w_2; m_3) = (3; 9471; 4305592257)$ and $d = 1$ is a solution with $V \approx 6,394$ and $\tau = 454,608\pi$. Whether there exist $V, \tau$ such that lattices with a maximum degree larger than 4 may exhibit fully product state permutation evolution is an open question.

The result for $D_{max} = 4$ required simultaneous solution of the equations for two different primes ($l = 2$ and $l = 3$) which suggests the conjecture that there are solutions to the system of equations for any $D_{max}$. Similar to the strategy above, by solving for $D_{max} = k$, it is possible

to construct a new Diophantine equation for $D_{max} = k + 1$. Determining whether this tower of equations is solvable is outside the scope of the present paper. On the other hand, as can already be seen in the case of $D_{max} = 4$, the values of $V, \tau$ for which the system exhibit such freezing for any initial number state quickly become prohibitively large for typical physical systems as the maximum lattice degree increases.

*Remark.* It is straightforward to generalize the Hamiltonian (23) to include more elaborate interactions as long as at each step the number operators associated with the neighbourhood of each evolving pair is constant. For example, we can write

$$\mathcal{H}_i = -t_{hop} \sum_{(i,j) \in A_i} (a_i^\dagger a_j + h.c.) + \sum_{i \in A_i} V_{ij} n_i n_j. \tag{32}$$

Given the number of particles in the neighborhood of each 2-site pair, we write (note here we include the potentials $V$ in the the definition of $\Delta$):

$$\Delta_{ij} = \sum_{k:\langle i,k \rangle} V_{ik} n_k - \sum_{k:\langle j,k \rangle} V_{jk} n_k, \tag{33}$$

and the freezing condition becomes:

$$\tau \sqrt{4 + \Delta_{ij}^2} = 2\pi m_{ij}, \quad m_{ij} \in \mathbb{Z}, \tag{34}$$

for all $\Delta_{ij}$ of the form (33).

## 2.6 Example 3: Deterministic evolution in the measurement induced chirality model on a Lieb lattice.

As another example, we consider the measurement induced chirality protocol of [54] with added nearest neighbour interactions and in the Zeno limit. In that work, a simple hopping Lieb lattice model of fermions was subjected to repeated measurements changing according to a prescribed chiral protocol. In contrast to the previous models, the Hamiltonian is not time dependent and all hopping terms in the Hamiltonian remain activated throughout the process.

It was shown in [54] that in the limit of rapid measurements, the so called the Zeno limit, the resulting dynamics is a classical stochastic process of permuting Fock states. We will see that, in this case too, we can find special values of interaction strength and protocol duration where the dynamics becomes deterministic. In fact, we will see the dynamics is governed by the same Diophantine equation as in example 2.

Specifically, we consider fermions hopping on a Lieb lattice with nearest-neighbor interactions given by

$$\mathcal{H} = -t_{hop} \sum_{<i,j>} a_i^\dagger a_j + V \sum_{<i,j>} n_i n_j. \tag{35}$$

We now apply the measurement protocol introduced in [54] to the system. Namely, we consider an 8 step measurement protocol in which, during the $i^{th}$ step that runs for a time $\tau$, the local particle density in all sites in a set $A_i^c$ of sites are measured. In the Zeno limit, all evolution during a step is restricted to neighboring sites in the subspace $A_i$ (See figure 4 for details), while the rest of the sites are kept frozen. Thus, in the Zeno limit, the evolution is effectively split into 8 steps evolved by the Hamiltonian (23), interspersed by an additional measurement. The measurements keep projecting the system onto Fock states, however, the particular states at hand are statistically distributed. However, if the step evolution (24) maps Fock states into Fock states, the whole procedure yields a deterministic evolution of an initial Fock state into another. In other words, the conditions for permutative evolution (and the

corresponding set of Diophantine equations) for this model are equivalent to those found in the interacting Floquet model investigated in example 2. This implies that the dynamics of the measurement induced chirality model and unitary Floquet evolution are equivalent at the special points in parameter space where Fock states are mapped to Fock states. However, if parameters are perturbed away from these special points, the dynamics of the two examples quickly begin to differ. This is due to the non-unitary nature of the measurements as opposed to the completely unitary evolution in the unitary Floquet case.

# 3 Hilbert Space Fragmentation

In Section 2.5, we gave $D_{\text{max}}$ conditions that must be simultaneously satisfied for Fock state permutative dynamics in models on a Lieb lattice with NN interactions. Similarly, in Section 2.1 we gave conditions for permutative evolution in the Floquet-Hubbard RLBL model. If in these models not all of these conditions are satisfied, then the evolution of a general initial state will require consideration of the full quantum many-body Floquet Hamiltonian.

However, evolution for certain initial states may still be deterministic even if only one or a few of the conditions for Fock state to Fock state evolution are met. This fragments [35] the Hilbert space, $\mathcal{H}$, into disconnected Krylov suspaces, $\mathcal{K}_i$, i.e.

$$\mathcal{H} = \bigoplus_i \mathcal{K}_i, \qquad \mathcal{K}_i = \text{span}_n\{\mathcal{U}^n|\psi_i\rangle\}, \tag{36}$$

where we have chosen a states $|\psi_i\rangle$ that are number local states in such a way that $\mathcal{K}_i$ are unique. In the rest of this section, we will explore the nature of the Hilbert space fragmentation in the example interacting Floquet and measurement induced models discussed in the previous section. Namely, we will see how the Hilbert spaces in these systems simultaneously support Krylov subspaces that are one-dimensional and correspond to frozen product states, few dimensional and correspond to states that evolve according to a classical cellular automation [47,48], and exponentially large subspaces that may evolve with more generic quantum many-body evolution.

## 3.1 Arrested development

Let us take as an example the NN-RLBL model on a Lieb lattice considered in Section 2.4. There, it was shown that when the parameters $V, \tau$ satisfy the conditions (26) and (27) with $m_0$ even, then the evolution of each step in the Floquet drive is given by the identity. However, certain initial states do not require all of the conditions (26) and (27) to be satisfied in order for this freezing of the dynamics to occur. For example, initial Fock states where $\Delta = 1$ for every activated 2-site pair only require $\sqrt{4 + V^2}\tau = 2\pi m_1$ with $m_1 \in \mathbb{Z}$ to be satisfied in order to exhibit frozen dynamics. Even if every other condition (26) and (27) with $\Delta \neq 1$ fails to be satisfied, such states will still be frozen under the NN-RLBL evolution. However, in this case, initial states containing at least one 2-site pair with $\Delta \neq 1$ may evolve into a superposition of Fock states. Therefore, the Hilbert space has been split (fragmented) into subspaces of Fock states that are frozen and a subspace of states which are not frozen.

In Fig. 6 we give examples of frozen particle configurations. At the top of the figure are configurations that require the satisfaction of only the $\Delta = 0$ condition (27) to be frozen, configurations in the middle of the figure require only the $\Delta = 1$ condition, and at the bottom of the figure is a particle configuration that will be frozen so long as both the $\Delta = 1$ and $\Delta = 3$ conditions (26) are satisfied. If the conditions for $\Delta = 0$, $\Delta = 1$, and $\Delta = 3$ are all satisfied then the entire Fig. 6 represents a frozen particle configuration. We emphasize here that,

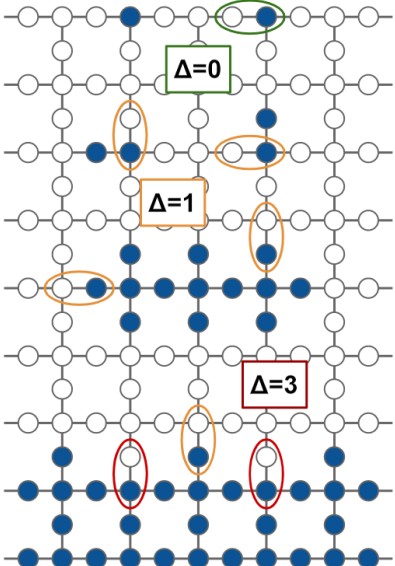

Figure 6: A Zoo of frozen particle configurations when only some of the conditions in (26) and (27) are satisfied on a nearest neighbour interacting Lieb-RLBL model. At the top, a particle configuration that requires only that the $\Delta = 0$ condition (and $m_0$ even) be satisfied for frozen evolution. In the bulk of the system are particle configurations that will be frozen so long as the $\Delta = 1$ condition is satisfied. The lower edge of the system provides an example of a particle configuration that will be frozen so long as both the $\Delta = 1$ and $\Delta = 3$ conditions are satisfied. Since all the particle configurations above are disconnected, the simultaneous satisfaction of the $\Delta = 0$, $\Delta = 1$, and $\Delta = 3$ conditions implies that the entire system above will be frozen.

even if only one of the conditions (27), (26) are satisfied, that the number of frozen particle configurations grows exponentially in system size.

Additionally, we note here that the chiral nature of the Floquet procedure played no role in the emergence of these frozen states. In fact, any procedure that sequentially activates hopping between neighboring pairs of sites (suitably spaced to keep evolution disjoint after adding NN interactions) will exhibit the exact same frozen states. For example, even if we consider a new procedure where, at each step in the evolution, the system is evolved with a $U_i$ from equation (22) chosen at random (uniformly), i.e. an example realization of this aperiodic, random evolution is given by

$$U = \ldots U_4 U_5 U_3 U_3 U_1 U_2 U_7 U_3 \,. \tag{37}$$

The exact same states will be frozen in this model as in the NN-RLBL model on a Lieb lattice.

Therefore, for any model of the form (37) we have the following situation. The Hilbert space is fragmented into a (exponentially large) non-frozen subspace and an exponential number of subspaces corresponding to frozen states. In general cases, initial states in the non-frozen subspace are free to ergodically explore their Krylov subspace leading to chaotic dynamics. Such behavior is referred to as Krylov-restricted thermalization [55]. However, if additional symmetries and structure are present, the non-frozen subspace may be further split into additional subspaces (see Sec. 3.4).

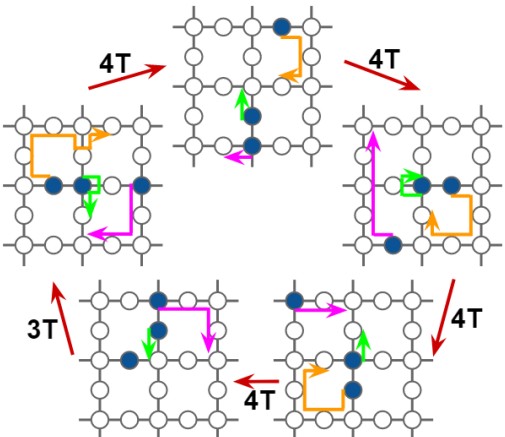

Figure 7: Example evolution within a cellular automation Krylov subspace set by the simultaneous satisfaction of the $\Delta = 0$ and $\Delta = 1$ conditions in equations (26) and (27). In this case, 2-site pairs with $\Delta = 0$ evolve with perfect swapping while 2-site pairs with $\Delta = 1$ are frozen. The resulting cellular automation for this example initial particle configuration results in the particles returning to their initial sites after $19T$. Example values of $V, \tau$ that achieve this evolution are $V = \sqrt{12}$ and $\tau = \frac{\pi}{2}$. Particle trajectories are drawn with orange, green, and magenta arrows.

## 3.2 Krylov Subspaces of Cellular Automation

Since the dynamics of a particle configuration that obey the Diophantine conditions depends crucially on particles on the neighbouring sites, it can be naturally encoded as a cellular automation step. We will now see how Krylov subspaces supporting classical CA [47, 48] at each evolution step may emerge in interacting Floquet and measurement-induced systems when a few of the conditions for number state to number state evolution are satisfied.

To elucidate this effect, we consider again the NN-RLBL model on the Lieb lattice. In this case, we take the $\Delta = 0$ and the $\Delta = 1$ conditions for number state to number state evolution to both be satisfied, but this time the $\Delta = 0$ condition is satisfied for perfect swapping while the $\Delta = 1$ condition is satisfied for freezing. This may happen at, for example, $\tau = \frac{\pi}{2}$ and $V = \sqrt{12}$.

It is now possible to find number states such that the initial particle configuration, $|\Psi_{init}\rangle$, and the resulting states after evolution of each step in the Floquet drive, all satisfy either $\Delta = 0$ or $\Delta = 1$ for every activated two-site pair in the system with a single particle. We give an example particle configuration where this may occur in Figure 7. Here, the space of states $span_n\{U^n|\Psi_{init}\rangle\}$ defines a Krylov subspace where evolution is completely given by a CA since at each step in the Floquet drive the local particle densities are updated deterministically based on the neighboring particle densities (i.e. if $\Delta = 0$ or 1).

Similarly to the case of frozen initial particle configurations, disjoint unions of particle configurations that evolve as a CA will also evolve as a CA. For particle configurations whose CA evolution leaves all particles contained in a volume that does not scale with system size (for example, the evolution of the configuration in Figure 7 remains contained within the $5 \times 5$ site square), the number of CA Krylov subspaces will grow exponentially with the system size (since there are exponentially many disjoint unions of such particle configurations). These CA subspaces may coexist with frozen Krylov subspaces as well as with exponentially large subspaces with more general quantum evolution.

It is important to note that these CA subspaces break the underlying $T$ time translation symmetry of the evolution operator. For example, the particle configuration in Fig. 7 returns to its initial configuration after $19T$. However, the exact realization of this Krylov subspace

requires fine-tuning in parameter space. If an alteration of this model was possible such that the realization of these Krylov subspaces did not require fine-tuning, then such a model would be a realization of a time-crystal. In fact, since the systems we've considered may simultaneously support Krylov suspaces that break the $T$ time translation symmetry in different ways, such a stabilized system would simultaneously support several different time crystals depending on which Krylov subspace contains the initial state. Recent works [49, 50] have argued that disorder may stabilize dynamics for regions in parameter space near similarly fine-tuned points in an interacting, RLBL model to achieve anomalous Floquet insulating phases. The basic idea is to consider a high frequency regime, where the expansion studied in e.g. [26, 56] shows that the effective evolution in the high frequency limit only acts non-trivially on small resonant spots that take an exponentially long time to destroy localization. This in turn is associated with the robustness of prethermal phases and localization when an MBL Floquet system is perturbed. In the RLBL system our special points for the model require, for example, $\tau \sim \frac{\pi}{2}$ which is not a high frequency drive. However, if the system is instead viewed in the rotating frame of the chiral RLBL drive, the evolution can be effectively presented as a high frequency drive, related to the inverse of the parameter offsets between $\tau, V$ and the perfect point (see [49] for details). It is thus possible to show that disorder stabilizes the evolution. In an upcoming work [53] we extend this treatment to address when disorder may stabilize dynamics for the entire system or for specific Krylov subspaces in a more general set of models.

### 3.3 Frozen states of Floquet evolution on a chain with nearest neighbour interactions

A major tool used in the analysis of the interacting Floquet and measurement models above was that the interactions preserved the disjoint nature of the steps of the periodic drive. However, using the same tools as in the disjoint case, it is possible to find frozen states even when the activated neighboring pairs interact (i.e. do not commute).

Here, we investigate an example model where the interactions ruin the disjoint nature of the Floquet drive and show how, at special values of interaction strength and driving frequency, it is still possible to find states that are frozen. Namely, we take as an example a 1D, NN interacting Hamiltonian of the form

$$\mathcal{H}(t) = H_0(t) + V \sum_{i=0}^{N-2} n_i n_{i+1}, \tag{38}$$

where

$$H_0(t) = \begin{cases} \sum_{i \text{ even}} (a_i^\dagger a_{i+1} + h.c.), & 0 \le t < \frac{T}{2} \equiv \tau, \\ \sum_{i \text{ odd}} (a_i^\dagger a_{i+1} + h.c.), & \frac{T}{2} \le t < T. \end{cases} \tag{39}$$

Similarly to the previous cases, let us again consider a single 2-site pair where hopping is activated. If the occupancy of the sites neighboring the pair happen to be static, then the conditions for frozen or perfect swapping (26) and (27) will still hold (except here with $D_{max} = 2$). However, this is, of course, not generally the case. Even if a neighboring pair is stroboscopically frozen, the number of particles at neighbouring sites may change during the evolution and ruin the conditions (26) and (27).

However, if every 2-site pair with a single particle is located on the edge of a domain wall in the system, then $\Delta$ will again be well defined (since any neighboring particles will be stationary due to Pauli exclusion) and the conditions (26) and (27) will hold for these particle configurations. In Figure 8, we give examples of such states that will be stroboscopically frozen when the $\Delta = 1$ condition is satisfied, i.e. all these states are eigenstates to the evolution operator $\mathcal{U}(T) = \mathcal{T}e^{-i\int_0^T \mathcal{H}(t)}$.

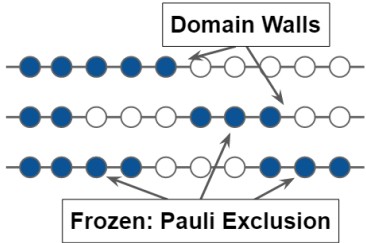

Figure 8: Particle configurations frozen in the Even-Odd NN model at values of $V, \tau$ that satisfy the $\Delta = 1$ condition in (26). The only 2-site pairs with a single particle are located on the domain walls. Since, within the uniform domain, particles are frozen at all times due to Pauli exclusion, the neighboring particle number difference for 2-site pairs on the domain wall is constant and given by $\Delta = 1$.

We now turn to numerically investigating the emergence of these frozen states and the Hilbert space fragmentation in this system. We exactly diagonalize $\mathcal{U}(T)$ at the special points $V = \sqrt{12}$, $\tau = \frac{\pi}{2}$ and $\tau = \pi$.[2] Here, the condition for frozen $\Delta = 1$ is satisfied, while $\Delta = 0$ is perfect swapping or frozen respectively. If the activated neighboring pairs were disjoint, evolution at these parameter values would be exactly solvable (with dynamics either being a CA or stroboscopically frozen). As we will see, however, this is not the case here. The Hilbert space instead fragments into exponentially many subspaces of frozen domain wall states and a single, exponentially large, ergodic subspace.

To seperate the two classes of subspaces, we calculate the half-chain entanglement entropy of the eigenstates (shown in Figure 9). The frozen eigenstates have zero entanglement entropy while the other eigenstates have finite (and as can be seen from Fig. 9, large) entanglement entropy. Upon plotting the average local particle densities of a sample of the zero entanglement entropy eigenstates, we find that they do indeed correspond to the expected frozen domain wall states.

While the number of domain wall states $\mathcal{N}_{\text{Froz}}$ grows exponentially with system size, their fraction of the total Hilbert space dimension goes to zero. To see this, consider e.g. states that satisfy the $\Delta = 1$ condition (with similar consideration applying to $\Delta = 0$ situations). Such states are characterized by occupied domains that are separated from each other by at least 3 sites (see Fig (8)) so that particles on the edges of separate domains do not interact with each other at any stage of the evolution. A rough lower bound on the number of such states is $2^{\lfloor N/3 \rfloor}$, if we only look at domains whose length is a multiple of 3 starting at sites $3, 6, ..., \lfloor N/3 \rfloor$. An upper bound can easily be obtained by noticing that the total number of domains cannot exceed $\lfloor N/3 \rfloor$, thus we have an upper bound by considering the entropy of the positions of domain wall boundaries $\sum_{k=0}^{\lfloor N/3 \rfloor} \binom{N}{k}$. In the large $N$ limit, this is dominated by the term $\binom{N}{N/3}$ which scales as $2^{N S_{\text{bi}}(1/3)} \approx 2^{0.92N}$, by Stirlings approximation, with the entropy function $S_{\text{bi}}(a) = a \log_2(\frac{1}{a}) + (1-a) \log_2(\frac{1}{1-a})$. In summary, we see that $\mathcal{N}_{\text{Froz}}$ grows exponentially in system size but the fraction of frozen states compared to the full Hilbert space dimension (which scales as $2^N$) is zero in the thermodynamic limit.

The large half-chain entanglement entropy of non-domain wall states suggests that the rest of the Hilbert space might be thermalized. To provide further evidence to this claim, we analyze an indicator often used to differentiate between ergodic and integrable systems: the statistics of level spacing ratios.

---

[2]As a technical note, the frozen domain wall states will be highly degenerate and numerical diagonalization will give a random basis of eigenstates within the degenerate subspace. To find the frozen states within this basis, we apply a small disorder potential during the wait step in the evolution to split the energy levels. This disorder potential will add only a global phase to the frozen states and thus allows a direct numerical route to finding them.

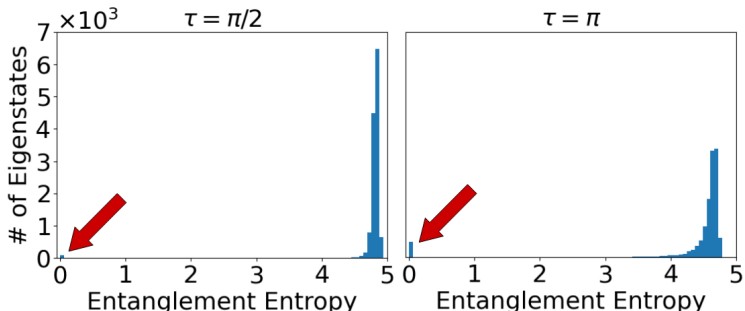

Figure 9: Half-chain entanglement entropy of all eigenstates of the evolution in the even-odd NN Floquet model. Eigenstates were found by exactly diagonalizing a 16 site chain. The parameter values were chosen such that both the $\Delta = 0$ and $\Delta = 1$ conditions in (26) and (27) are satisfied: $V = \sqrt{12}$, $\tau = \frac{\pi}{2}$ (left) and $\tau = \pi$ (right). Despite the non-disjoint nature of the activated hopping site pairs, the conditions (26) and (27) will still be valid for domain wall states that will, therefore, be frozen under the dynamics. These number states have no entanglement entropy and are indicated with red arrows in the figure above. The other eigenstates exhibit near-maximal entanglement entropy. This is a signature of the fragmentation of the Hilbert space into frozen Krylov subspaces and a ergodic Krylov subspace.

For thermalizing systems, it is expected [14] that the evolution operator $\mathcal{U}$ resembles random matrices drawn from a circular ensemble (the analog of gaussian ensembles for unitary matrices). Unlike the evolution operators for integrable systems, eigenstates of circular ensembles are random vectors and the spectrum exhibits level repulsion. Thus, it is possible to argue whether a system is ergodic by analyzing the statistics of the spacing of energy levels to see if the distribution is Poissonian (corresponding to no level repulsion) or if it corresponds to the expected level spacing distribution of circular ensembles (see [14] for explicit formulas).

Namely, consider the level spacings between two neighboring eigen-quasienergies $\varepsilon$ (i.e. $\varepsilon$'s are the phases of the eigenvalues of $\mathcal{U}$),

$$\delta_n = \varepsilon_{n+1} - \varepsilon_n. \tag{40}$$

The ratio of level spacings is given by

$$r_n = \frac{min\{\delta_n, \delta_{n+1}\}}{max\{\delta_n, \delta_{n+1}\}}. \tag{41}$$

We then expect the statistics of $r$ to match that of the circular ensembles instead of yielding a Poissonian distribution if the system is ergodic.

In our case, however, the system is not completely ergodic since the domain wall number states are eigenstates of the evolution. We instead wish to study the nature of the subspace which is the compliment of the set of all frozen Krylov subspaces within the Hilbert space. We thus will only consider $\delta_n$ in (40) if the corresponding eigenstates of $\varepsilon_{n+1}$ and $\varepsilon_n$ have non-zero half-chain entanglement entropy. The results of this analysis are shown in Fig. 10. As can be seen in the figure, the probability distribution is in good agreement with that of the circular orthogonal ensemble (COE) suggesting that the Krylov subspace is thermal.

In summary, we have shown that the Hilbert space of the even-odd NN Floquet model is fragmented at special values of interaction strength and driving frequency. The fragmented Hilbert space simultaneously supports exponentially many (in system size) frozen Krylov subspaces and a single, exponentially large ergodic Krylov subspace. In this model, we did not find evidence of CA subspaces. Whether these subspaces are realizable in other non-disjoint

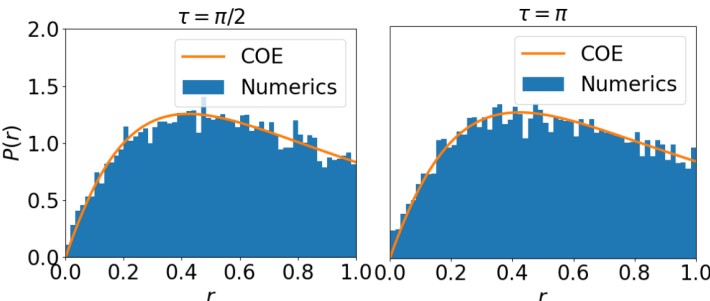

Figure 10: Level spacing statistics in the non-frozen Krylov subspace for evolution in the even-odd NN Floquet model. As in Fig. 9, parameter values are chosen as $V = \sqrt{12}$, $\tau = \frac{\pi}{2}$ (left) and $\tau = \pi$ (right). The probability distribution, $P(r)$, of the level spacing ratios, $r$, for quasi-energy levels not corresponding to frozen eigenstates provides good agreement with the level spacing probability distribution of random matrices in the circular orthogonal ensemble (COE). This suggests that the Krylov subspace is ergodic.

models is an open question. Furthermore, for neighboring two-site pairs each with a single particle, the interactions between the pairs could conspire to produce special values of $V, \tau$ not given by equations (26) and (27) where evolution is stroboscopically frozen. We leave both these open questions for future work.

## 3.4 Remarks on Fragmentation and Ergodicity in NN-RLBL

In this section, we study numerically the entanglement entropy of eigenstates and level statistics in the NN-RLBL model with the Diophantine conditions only partly satisfied. We show that the fragmentation seems to allow for a yet richer structure than shown in the 1D example above. In particular, we consider the NN-RLBL model with parameters $\tau = \frac{2\pi}{\sqrt{7}}$ and $V = \sqrt{3}$. Here, activated hopping pairs with $\Delta = 1$ are frozen, but none of the other conditions (26) or (27) are satisfied. We plot the entanglement entropy of the eigenstates of the evolution in Fig. 11.

Examining Fig. 11 we find that there are frozen particle configurations leading to eigenstates with no entanglement entropy. There are no cellular automation subspaces since only a Diophantine condition for freezing has been satisfied. Additionally, there is a subspace containing eigenstates with near-maximal entanglement entropy suggesting ergodicity. However, there is also a subspace of eigenstates with entanglement entropy that is neither zero nor maximal. This suggests that the subspace compliment to the frozen and cellular automation subspaces has been further fragmented into ergodic and non-ergodic subspaces.

We further investigate this phenomena by examining the level spacing statistics for the non-frozen subspace in Fig. 12. The statistics show some level repulsion as they do not match that of the Poisson distribution, but the sampled ratios also do not align with the distribution associated with the relevant ensemble of random matrices, the circular unitary ensamble (CUE).

We now further separate the non-frozen subspace into the subspace spanned by the eigenstates with near-maximal entanglement entropy and the subspace spanned by the eigenstates without near-maximal entanglement entropy. Fig. 12 shows that the level spacing statistics corresponding to the near-maximal states agrees with the distribution for the associated random matrix ensemble, while the level spacing statistics associated with the eigenstates without near-maximal entanglement entropy seems to exhibit Poisson statistics. This, therefore, provides evidence to suggest that the non-frozen subspace has fragmented into a subspace exhibiting chaotic dynamics and a subspace exhibiting integrable dynamics.

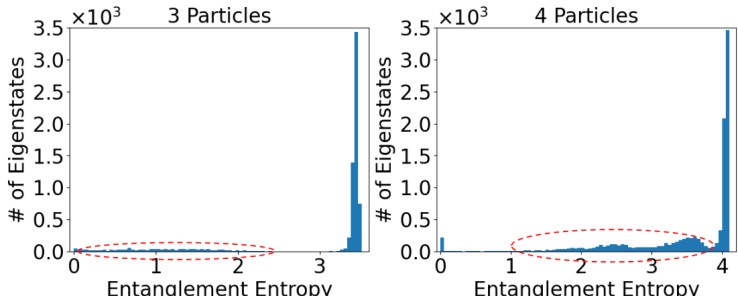

Figure 11: Entanglement entropy for eigenstates of the evolution in the NN-RLBL model. Parameters are given by $\tau = \frac{2\pi}{\sqrt{7}}$ and $V = \sqrt{3}$ which corresponds to activated hopping pairs with $\Delta = 1$ frozen and none of the other $\Delta$ conditions satisfied. On the left we consider evolution of 3 particles in an $8 \times 6$ lattice and on the right the evolution of 4 particles in an $8 \times 4$ lattice. In both cases, $\Delta = 1$ frozen yields frozen particle configurations leading to a set of zero entanglement entropy eigenstates. Additionally, there is a subspace of near-maximal entanglement entropy eigenstates. However, circled in red, there are eigenstates with entanglement entropy in-between the two extreme values. The span of these eigenstates forms another non-ergodic subspace of the Hilbert space that is distinct from frozen or cellular automation subspaces.

We suspect that the splitting of the non-frozen subspace may be further understood via a close consideration of the symmetries of the Floquet model, the satisfied Diophantine conditions, and their interplay. Similarly, it may be possible to describe in more detail the dynamics of the putative integrable subspace. However, we leave such investigations for future work.

## 4    Summary and discussion

In recent years the study of quantum many body states that break ergodicity has been an active field of research. Here, we considered conditions for dynamics in interacting systems that takes initial local number states to local number states. We have found such conditions for systems with sequentially activated hopping involving interactions such as Hubbard and nearest neighbour density interactions. Studying the resultant Diophantine relations between interaction strength, hopping energy, and hopping activation time, we discovered solutions to a variety of such systems. The resultant dynamics can be cast into two types: (1) Evolution that is deterministic for any initial Fock state (2) Fragmentation of the Hilbert space into deterministic sub-spaces and non-deterministic ones.

Our results introduce new sets of dynamically tractable interacting systems, with an emphasis on 2d where such results are scarce. Furthermore, the approach is applicable to similar systems in other dimensions. At the special solvable points, we get a variaty of behaviors from frozen dynamics of Fock states to cellular automata like evolution of selected subspaces. In cases where only some of the Diophantine conditions are met, we have shown that the special subspaces can exist simultaneously with states that possess volume law entanglement entropy and level statistics suggesting thermalizing behavior.

As discussed in section 3.2, although the ratios of Hamiltonian parameters (interaction strength, evolution time etc) considered here are finely tuned, previous work suggests that similarly finely tuned points may be stabilized by disorder to realize novel dynamical phases. In particular, periodic celullar automata evolution in our models may lead to new classes of time crystals.

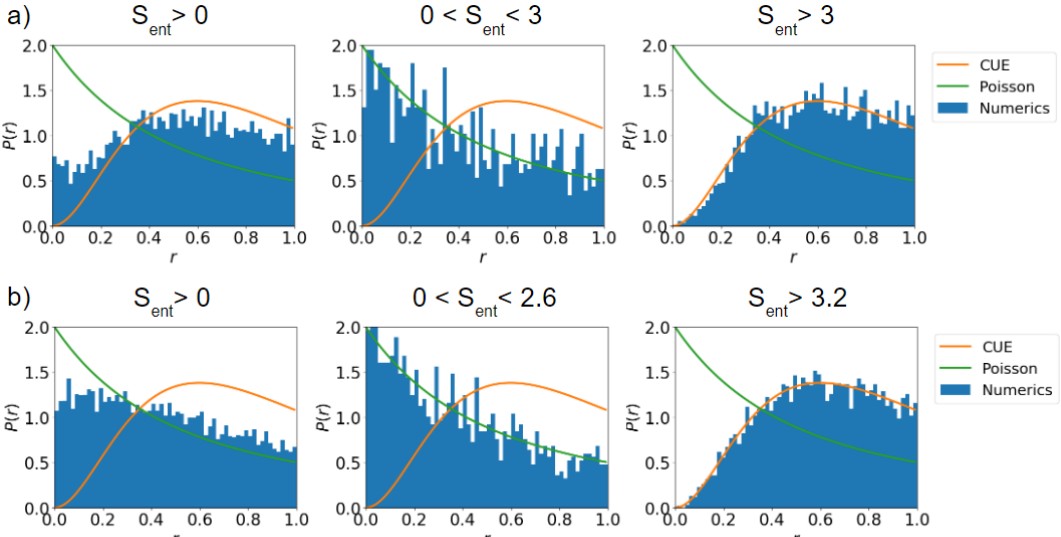

Figure 12: Level spacing statistics in the NN-RLBL model. Similar to Fig. 11, we consider evolution at $\tau = \frac{2\pi}{\sqrt{7}}$ and $V = \sqrt{3}$ with two different lattice sizes and fillings: a) 3 particles in an $8 \times 6$ lattice, b) 4 particles in an $8 \times 4$ lattice. In both cases, we have, from left to right: level spacing statistics of the full Hilbert space minus the frozen subspace, statistics for non-frozen eigenstates with entanglement entropy, $S_{\text{ent}}$, less than a cutoff value [3 for a) and 2.6 for b)], and statistics for eigenstates with entanglement entropy greater than a cutoff [3 for a) and 3.2 for b)]. The level spacing statistics for all non-frozen eigenstates is a mixture of Poisson statistics (corresponding to the subspace spanned by the eigenstates with $S_{\text{ent}}$ less than a given cutoff) and circular unitary ensemble (CUE) statistics (corresponding to the subspace spanned by the eigenstates with $S_{\text{ent}}$ more than a given cutoff). In the 4 particle case, the entanglement entropy range between 2.6 and 3.2 contains a mixture of Poisson and CUE statistics, so we focus on the eigenstates with $S_{\text{ent}}$ less than 2.6 or more than 3.2 where the Poisson and CUE subspaces may be easily distinguished.

The problem of finding complete freezing of Fock states also led us to an interesting number theoretic problem involving the solution of a tower of Diophantine equations described in (26) and (27). We have shown explicitly solutions for dynamics on lattices with maximal degree of up to 4 nearest neighbours and conjecture a solution can be found for arbitrary maximal degree.

Another interesting question is that of the effect of long range interactions. We expect that two main consequences may arise: (1) The removal of all exactly solvable points, possibly destroying classical cellular automata like behavior in the system and (2) The long-range interactions may act as an effective disorder (due to the many possible configurations of particles away from particular activated sites where the dynamics occurs) stabilizing the classical like dynamics. A further possibility is that interactions with finely tuned decay properties may facilitate additional Diophantine conditions. However, except in rare cases, we do not expect that the new equations will combine into a solvable system.

Finally, we remark that the Diophantine methods utilized in this work may be applicable to bosonic systems, and systems with pairing terms where resultant cellular automata may not be of the number preserving type.

## Acknowledgements

We thank G. Refael and E. Berg for discussions.

**Funding information** Our work was supported in part by the NSF grant DMR-1918207.

## A Solutions for subspaces of two sites

### A.1 Hubbard Floquet Evolution of 2-site Pair in the 2-particle Sector

We index the 4-particle configurations of the subspace as follows:

$$0 \to \ \uparrow\downarrow \ \ \underline{\ \ }, \tag{A.1a}$$

$$1 \to \ \underline{\ \ } \ \ \uparrow\downarrow, \tag{A.1b}$$

$$2 \to \ \uparrow_{\underline{\ }} \ \ \underline{\ }\downarrow, \tag{A.1c}$$

$$3 \to \ \underline{\ }\downarrow \ \ \uparrow_{\underline{\ }}. \tag{A.1d}$$

We therefore have that the representation of the Hubbard Hamiltonian (2) in this subspace is given by

$$\mathcal{H} = \begin{pmatrix} V & 0 & -1 & -1 \\ 0 & V & -1 & -1 \\ -1 & -1 & 0 & 0 \\ -1 & -1 & 0 & 0 \end{pmatrix}. \tag{A.2}$$

Hence, the evolution, $\mathcal{U} = e^{-i\mathcal{H}\tau}$, is given by

$$\mathcal{U} = e^{-\frac{1}{2}iV\tau} \begin{pmatrix} e^{-\frac{1}{2}iV\tau}\left[\frac{1}{2}+A\right] & e^{-\frac{1}{2}iV\tau}\left[-\frac{1}{2}+A\right] & B & B \\ e^{-\frac{1}{2}iV\tau}\left[-\frac{1}{2}+A\right] & e^{-\frac{1}{2}iV\tau}\left[\frac{1}{2}+A\right] & B & B \\ B & B & e^{\frac{1}{2}iV\tau}\left[\frac{1}{2}+\bar{A}\right] & e^{\frac{1}{2}iV\tau}\left[-\frac{1}{2}+\bar{A}\right] \\ B & B & e^{\frac{1}{2}iV\tau}\left[-\frac{1}{2}+\bar{A}\right] & e^{\frac{1}{2}iV\tau}\left[\frac{1}{2}+\bar{A}\right] \end{pmatrix}, \tag{A.3}$$

where

$$A(V,\tau) = \frac{e^{\frac{1}{2}iV\tau}}{2}\left[\cos(\frac{1}{2}\tau\sqrt{16+V^2}) - i\frac{V}{\sqrt{16+V^2}}\sin(\frac{1}{2}\tau\sqrt{16+V^2})\right], \tag{A.4}$$

$$B(V,\tau) = 2i\frac{\sin(\frac{1}{2}\tau\sqrt{16+V^2})}{\sqrt{16+V^2}}, \tag{A.5}$$

and $\bar{A}$ is the complex conjugate.

We are now interested in finding when (A.3) is a permutation matrix. Note, for non-zero $V$, $|B| < 1$. Thus, our only hope for a permutation matrix is if $B = 0$. This occurs when $\frac{1}{2}\tau\sqrt{16+V^2} = \pi m$ for some $m \in \mathbb{Z}$, i.e. the condition given in (5).

Solving for $A(V,\tau)$ at condition (5) yields

$$A(V,\tau)|_{Condition:(5)} = \frac{1}{2}e^{i[\pi m + \frac{1}{2}V\tau]}. \tag{A.6}$$

In (A.3), $\mathcal{U}$ is a permutation matrix when, in addition to the requirement $B = 0$, $|A| = \frac{1}{2}$ and $\frac{V\tau}{\pi} \in \mathbb{Z}$. These 2 conditions are uniquely met when, using (A.6), $\pi m + \frac{1}{2}V\tau = \pi n$ for some $n \in \mathbb{Z}$. Thus, we have arrived at the condition given in (6). When (5) and (6) are satisfied, $\mathcal{U}$ then becomes

$$\mathcal{U}|_{\text{Conditions: (5) and (6)}} = \begin{pmatrix} n-1 & n & 0 & 0 \\ n & n-1 & 0 & 0 \\ 0 & 0 & n-1 & n \\ 0 & 0 & n & n-1 \end{pmatrix} \mod 2, \qquad (A.7)$$

i.e. yielding the result that when $n$ is even (odd) evolution is the identity (perfect swapping).

## A.2  Nearest Neighbor Floquet Evolution of 2-site Pair in 1-particle Sector

The Hamiltonian of the $j^{th}$ 2-site pair is

$$\mathcal{H}_j = -(a_{j1}^\dagger a_{j2} + h.c.) + V n_{j1} n_{j2} + V N_1 n_{j1} + V N_2 n_{j2}, \qquad (A.8)$$

where $N_1, N_2$ correspond to the number of particles (outside the $j^{th}$ pair) neighboring site 1 and site 2 in pair $j$ respectively. Note, $[N_1, \mathcal{H}_j] = [N_2, \mathcal{H}_j] = 0$.

The representation of the Nearest Neighbor Hamiltonian in the 1-particle sector is given by

$$\mathcal{H} = \begin{pmatrix} N_1 V & -1 \\ -1 & N_2 V \end{pmatrix}. \qquad (A.9)$$

Hence, the evolution, $\mathcal{U} = e^{-i\mathcal{H}\tau}$, is given by

$$\mathcal{U} = \frac{e^{-\frac{1}{2}i(N_1+N_2)V\tau}}{C} \begin{pmatrix} C\cos\left[\frac{C}{2}\tau\right] - i\Delta V \sin\left[\frac{C}{2}\tau\right] & 2i\sin\left[\frac{C}{2}\tau\right] \\ 2i\sin\left[\frac{C}{2}\tau\right] & C\cos\left[\frac{C}{2}\tau\right] + i\Delta V \sin\left[\frac{C}{2}\tau\right] \end{pmatrix}, \quad (A.10)$$

where

$$C(\Delta, V) \equiv \sqrt{4 + \Delta^2 V^2}, \qquad (A.11)$$

$$\Delta \equiv N_1 - N_2. \qquad (A.12)$$

For perfect swapping to occur, we must have that the diagonal elements of (A.10) go to zero. This may only occur when

$$\frac{C}{2}\tau = \pi\left(m + \frac{1}{2}\right), \quad \text{for} \quad m \in \mathbb{Z}, \quad \Delta = 0. \qquad (A.13)$$

For freezing to occur, we must have that the off-diagonal elements of (A.10) are zero. Note, depending on the particle configuration, $\Delta$ may take any value such that

$$\Delta \in \mathbb{Z}, \quad \text{and} \quad |\Delta| < \max[\deg(\text{site 1}), \deg(\text{site 2})],$$

where deg is the degree of the vertex. We must therefore have that $\frac{C}{2}\tau = \pi m$ for all possible values of $\Delta$ and with $m \in \mathbb{Z}$, i.e. letting $\Delta_i \in \{0, 1, ..., \max[\deg(\text{site 1}), \deg(\text{site 2})] - 1\}$ such that $\Delta_i = \Delta_j$ iff $i = j$, we require

$$\frac{C(\Delta_i, V)}{2}\tau = \pi m_i, \quad \forall i, \qquad (A.14)$$

where $m_i \in \mathbb{Z}$.

Combining Equations (A.13) and (A.14) yields (26) and (27) in the main text.

We may now proceed by solving one value of $\Delta_i$ at a time. We start with $\Delta_0$ and, without loss of generality, let $\Delta_0 \neq 0$ (if $\Delta_0 = 0$, we may simply replace $m_0 \to \frac{m_0}{2}$ in the final result), we have from (A.14) that

$$\tau = \frac{2\pi m_0}{\sqrt{4 + \Delta_0^2 V^2}}. \tag{A.15}$$

Now, looking next at $\Delta_1 \neq 0$ (again, we may set $m_1 \to \frac{m_1}{2}$ if $\Delta_1 = 0$), we use Eqs. (A.14) and (A.15) to find

$$\frac{\sqrt{4 + \Delta_1^2 V^2}}{\sqrt{4 + \Delta_0^2 V^2}} \pi m_0 = \pi m_1 \implies V = 2\sqrt{\frac{m_1^2 - m_0^2}{\Delta_1^2 m_0^2 - \Delta_0^2 m_1^2}}. \tag{A.16}$$

Now, taking any $\Delta_j$ such that $j \geq 2$ and combining Eqs. (A.14), (A.15), and (A.16) and simplifying we find

$$m_0^2(\Delta_1^2 - \Delta_j^2) + m_1^2(\Delta_j^2 - \Delta_0^2) + m_j^2(\Delta_0^2 - \Delta_1^2) = 0, \qquad m_i \to \frac{m_i}{2}, \quad \text{if} \Delta_i = 0. \tag{A.17}$$

Equation (A.17) therefore corresponds to a set of $\max[\deg(\text{site } 1), \deg(\text{site } 2)] - 3$ Diophantine equations that must be solved simultaneously to find the values of $m_i$ (and thus $V, \tau$) that correspond to CA dynamics. Note, also, that in (A.17) we must replace $m_i \to \frac{m_i}{2}$ for whichever $\Delta_i = 0$.

Note, a particular solution for the first equation in (A.17) when $\Delta_0, \Delta_1, \Delta_2 \neq 0$ is $m_0 = \Delta_0$, $m_1 = \Delta_1$, $m_2 = \Delta_2$. Hence, using (14), the solution to (A.17) for $j = 2$ is given by

$$\begin{pmatrix} m_0 \\ m_1 \\ m_2 \end{pmatrix} = d \begin{pmatrix} -\left[(\Delta_1^2 - \Delta_2^2)w_1^2 - (\Delta_2^2 - \Delta_0^2)w_2^2\right]\Delta_0 - 2(\Delta_2^2 - \Delta_0^2)w_1 w_2 \Delta_1 \\ \left[(\Delta_1^2 - \Delta_2^2)w_1^2 - (\Delta_2^2 - \Delta_0^2)w_2^2\right]\Delta_1 - 2(\Delta_1^2 - \Delta_2^2)w_1 w_2 \Delta_0 \\ \left[(\Delta_1^2 - \Delta_2^2)w_1^2 + (\Delta_2^2 - \Delta_0^2)w_2^2\right]\Delta_2 \end{pmatrix}. \tag{A.18}$$

To obtain the equivalent of (A.18) when, for example, $\Delta_0 = 0$, we must take $m_0 = 2\Delta_0$ instead of $m_0 = \Delta_0$ in the particular solution of (A.17) and relatedly must use $A = \frac{\Delta_1^2 - \Delta_2^2}{4}$ instead of $A = \Delta_1^2 - \Delta_2^2$ in (14). Similar adjustments must be made to $B, m_1$ or $C, m_2$ if $\Delta_1 = 0$ or $\Delta_2 = 0$ respectively.

Equation (A.18) provides all possible solutions for $m_0, m_1, m_2$ (and thus $V, \tau$) that yield classical dynamics for any two site pair with $\Delta = \Delta_0, \Delta_1$, or $\Delta_2$. As a corollary, this implies that there exist values of $V, \tau$ (beyond the trivial $V = 0$ or $\tau = 0$ solutions) for any measurement protocol that sequentially isolates pairs of sites on a lattice such that all dynamics is a CA so long as the maximum degree of the lattice is at most 3. In other words, in this case, we may choose $\Delta_0 = 0$, $\Delta_1 = 1$, and $\Delta_2 = 2$ and (remembering to make the appropriate substitutions since $\Delta_0 = 0$) we find (A.18) becomes (30). As discussed in the main text, combining (30) with the $\Delta = 3$ condition yields a new Diophantine equation that may be solved numerically to find non-trivial solutions. For arbitrary maximal degree, a tower of Diophantine equations emerges. Whether solutions exist to these Diophantine equations for arbitrary maximal degree (and, if they exist, what they are) we leave as an open problem.

Below Eq. (30) we also made the comment that, in this case, $m_0$ is always even implying frozen dynamics. We here show that this is the case by way of contradiction. If $m_0$ is odd, then (30) implies

$$\frac{-32 w_1 w_2}{\gcd[-32 w_1 w_2, -3 w_1^2 - 16 w_2^2, -6 w_1^2 + 32 w_2^2]} \in 2\mathbb{Z} + 1, \tag{A.19}$$

i.e., an odd number. Hence, the gcd in the denominator must cancel all the powers of 2 in $-32w_1 w_2$. This implies $w_1$ must be even, since otherwise $-3w_1^2 - 16w_2^2$ would be odd leaving the gcd in the denominator odd. Since $w_1$ and $w_2$ are coprime, $w_2$ must be odd. We factor out the powers of 2 from $w_1$ leaving $w_1 = 2^{a_1} w_1'$ where $w_1'$ is odd. In the following, we will use two properties of the gcd:

$$\gcd(A, B) = \gcd(A, B + mA), \tag{A.20}$$

$$\gcd(A_1 A_2, B) = \gcd(A_1, B)\gcd(A_2, B), \tag{A.21}$$

for $m \in \mathbb{Z}$ and $A_1, A_2$ coprime. We now get

$$\gcd[-32w_1 w_2, -3w_1^2 - 16w_2^2, -6w_1^2 + 32w_2^2] \tag{A.22}$$

$$= \gcd[-(2^{5+a_1})w_1' w_2, -3(2^{2a_1})w_1'^2 - 2^4 w_2^2, -3(2^{2a_1+1})w_1'^2 + 2^5 w_2^2] \tag{A.23}$$

$$= \gcd[-(2^{5+a_1})w_1' w_2, -3(2^{2a_1})w_1'^2 - 2^4 w_2^2, -3(2^{2a_1+2})w_1'^2] \tag{A.24}$$

$$= \gcd[(2^{5+a_1}), -3(2^{2a_1})w_1'^2 - 2^4 w_2^2, (2^{2a_1+2})]A, \tag{A.25}$$

where in the third line we have added twice the second term in (A.23) to the third term using (A.20) and in the fourth line we focus only on terms in the gcd that may contribute a factor of 2, compiling the rest of the terms (through the use of (A.21)) in $A$ given by

$$A = \gcd\left[-w_1' w_2, -3(2^{2a_1})w_1'^2 - 2^4 w_2^2, -3(2^{2a_1+2})w_1'^2\right]\gcd\left[(2^{5+a_1}), -3(2^{2a_1})w_1'^2 - 2^4 w_2^2, -3w_1'^2\right].$$

For the gcd (A.25) to cancel all the powers of 2 in $-32w_1 w_2$ (thus making (A.19) odd), we therefore must have that

$$2^{5+a_1} = \gcd\left[(2^{5+a_1}), -3(2^{2a_1})w_1'^2 - 2^4 w_2^2, (2^{2a_1+2})\right]. \tag{A.26}$$

This, however, is a contradiction since for $\gcd[(2^{5+a_1}), (2^{2a_1+2})] = 2^{5+a_1}$ we must have $a_1 \geq 3$, but then we have

$$\gcd[(2^{5+a_1}), -3(2^{2a_1})w_1'^2 - 2^4 w_2^2, (2^{2a_1+2})] = 2^4 \neq 2^{5+a_1}. \tag{A.27}$$

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
