# Peer review of "Arrested Development and Fragmentation in Strongly-Interacting Floquet Systems"

_SciPost Physics, doi:SciPost Phys. 14, 145 (2023)_

## Round 1 · Referee Report · Anonymous (Referee 2) · 2022-11-7

Report

In this manuscript, the authors examine various instances of interacting fermionic models with subsequently activated hopping, and derive the conditions on the parameters under which the evolution does not generate entanglement, both in one- and in two-dimensional systems. In these cases the evolution can be seen as a permutation among Fock's states. The authors find that such conditions can be expressed in terms of a set of Diophantine equations, which, in most cases, are analytically solvable. A weaker set of conditions guarantees that the evolution does not create entanglement within a subspace of the whole Hilbert space, leading to a fragmentation of the latter. It is shown how, in correspondence of these "special points", the system may exhibit peculiar behaviors: frozen (stroboscopic) dynamics, the emergence quantum cellular automata or the spontaneous breaking of the discrete time-translational Floquet symmetry. While these phenomena only take place in correspondence of fine-tuned value of the parameters, the authors suggest that they can be made robust by the presence of noise.

In my opinion the manuscript is well-written, while the results obtained are interesting and can guide a further research on the subject, even if their applicability is hindered by the necessity of fine-tuning in the couplings of the models. I think the manuscript deserves to be published in Physical Review B after addressing some minor issues.

Requested changes

1. On several occasions the authors cite Ref. 48 and Ref. 49 to substatiate their claim that the addition of disorder is expected to stabilize the special point's dynamics. I find that the readability and the self-consistency of the manuscript would be improved if the author summarized briefly such arguments.

2. While the possibility of Floquet Discrete Time Crystals in presence of disorder has been long investigated, it is known that the presence of long-range interactions can have a stabilizing effect as well. Do you think this could be the case also for this kind of systems, or the finite range of the coupling is crucial in order to find such "special points"?

3. Referring to Fig. 8, it looks like the "entanglement-free" eigenstates, while exponentially growing, represent however a vanishing fraction (in the thermodynamic limit) of the whole Hilbert space. Is this correct? It is possible to give an estimate of the number of such states for a fixed chain length?

4. Some nugae

A) In the fourth line of pag. 4, the time-parameter \tau is set to \pi/2, wile it is not yet specified that we are working in units were t_{top} = 1.

B) In the first equation after (56) (pag. 20), the notation "deg()" is introduced without being properly defined within the Appendix.

  • validity: -
  • significance: -
  • originality: -
  • clarity: -
  • formatting: -
  • grammar: -

Author:  Matthew Wampler  on 2023-01-23  [id 3263]

(in reply to Report 2 on 2022-11-07)

We thank the referee for a close reading of our manuscript and below give a point-by-point response to their remarks.

Referee: In this manuscript, the authors examine various instances of interacting fermionic models with subsequently activated hopping, and derive the conditions on the parameters under which the evolution does not generate entanglement, both in one- and in two-dimensional systems. In these cases the evolution can be seen as a permutation among Fock's states. The authors find that such conditions can be expressed in terms of a set of Diophantine equations, which, in most cases, are analytically solvable. A weaker set of conditions guarantees that the evolution does not create entanglement within a subspace of the whole Hilbert space, leading to a fragmentation of the latter. It is shown how, in correspondence of these "special points", the system may exhibit peculiar behaviors: frozen (stroboscopic) dynamics, the emergence quantum cellular automata or the spontaneous breaking of the discrete time-translational Floquet symmetry. While these phenomena only take place in correspondence of fine-tuned values of the parameters, the authors suggest that they can be made robust by the presence of noise. In my opinion the manuscript is well-written, while the results obtained are interesting and can guide a further research on the subject, even if their applicability is hindered by the necessity of fine-tuning in the couplings of the models. I think the manuscript deserves to be published in Physical Review B after addressing some minor issues.

Authors: We thank the referee for their positive remarks and recommendation for publication after minor changes.

Referee: 1. On several occasions the authors cite Ref. 48 and Ref. 49 to substantiate their claim that the addition of disorder is expected to stabilize the special point's dynamics. I find that the readability and the self-consistency of the manuscript would be improved if the author summarized briefly such arguments.

Authors: We have added a brief summary of the arguments used in Refs. 49 and 50 (previously 48 and 49) at the end of section 3.2 in our manuscript.

Referee: 2. While the possibility of Floquet Discrete Time Crystals in presence of disorder has been long investigated, it is known that the presence of long-range interactions can have a stabilizing effect as well. Do you think this could be the case also for this kind of systems, or the finite range of the coupling is crucial in order to find such "special points"?

Authors: This is an interesting question. One can distinguish between two possible effects of the long range interactions in the present context.

We have added the following to the discussion section: “Another interesting question is that of the effect of long range interactions. We expect that two main consequences may arise: (1) The removal of all exactly solvable points, possibly destroying classical cellular automata like behavior in the system and (2) The long-range interactions may act as an effective disorder (due to the many possible configurations of particles away from particular activated sites where the dynamics occurs) stabilizing the classical like dynamics. A further possibility is that interactions with finely tuned decay properties may facilitate additional Diophantine conditions. However, except in rare cases, we do not expect that the new equations will combine into a solvable system. “

Referee: 3. Referring to Fig. 8, it looks like the "entanglement-free" eigenstates, while exponentially growing, represent however a vanishing fraction (in the thermodynamic limit) of the whole Hilbert space. Is this correct? It is possible to give an estimate of the number of such states for a fixed chain length?

Authors: The referee is indeed correct. To clarify this we have added in the text an upper and lower bound on the number of Delta=1 states that shows that the number of entanglement free states grows exponentially with at minimum N/3 and at maximum 0.92 N (vs, the Hilber space dimension that grows as 2^N), hence while the lower bound shows that the growth is exponential, the upper bound shows that the fraction of such states goes to zero.

Referee: 4. Some nugae A) In the fourth line of pag. 4, the time-parameter \tau is set to \pi/2, wile it is not yet specified that we are working in units were t_{top} = 1. B) In the first equation after (56) (pag. 20), the notation "deg()" is introduced without being properly defined within the Appendix.

Authors: We have corrected the above nugae.

Anonymous on 2023-01-30  [id 3281]

(in reply to Matthew Wampler on 2023-01-23 [id 3263])

In my opinion the authors have satisfyingly addressed all my remarks. I therefore recommend the publication of the manuscript.

---

## Round 1 · Referee Report · Anonymous (Referee 1) · 2022-11-7

Strengths

- timely topic
- experimentally relevant models

Weaknesses

- the absence of numerics
- the findings are not sufficiently evidenced
- not clearly written

Report

This work explores a form of ergodicity breaking, stemming from specific conditions on Floquet dynamics for fermionic systems.

The paper focuses on two examples of two-dimensional Floquet systems, where the dynamics are induced by sequentially activating hoppings between the pairs. By looking for the conditions for which the dynamics are frozen, the authors find Diophantine equations which admit some solutions and focus on those regimes.
After writing down the conditions for the examples under analysis, the authors discuss the situation in which only some of these conditions are met, which shall result in Hilbert space fragmentation.

The paper addresses a timely topic - currently the focus of great attention - and considers two-dimensional models with Floquet dynamics, which have sensible experimental applications. Nevertheless, the paper does not provide sufficient evidence to substantiate its claims and therefore I do not feel to recommend publication in the present form.

Let me be more specific. Despite the authors discussing in great detail the examples of the Hubbard-RLBL dynamics and the nearest neighbour's interactions on the Lieb lattice, both from the frozen dynamics and the fragmentation, no numerical evidence is provided on these two examples, but only pictorial representations of the evolution. The only numerical analysis (from which the authors deduce plenty of consequences) is provided for only one system size and leaves a lot of open questions: what would be the result of other local observables? How does the size of the two Krylov subspace scale with N?

Some other comments:
- the authors conjecture the existence of solutions for Dmax generic, because a solution is found for Dmax=4. What are the other arguments for this claim?
- it is not clear what is the difference between Example 2 and Example 3 (studied in Ref. [52] given by nearest neighbour interactions + measurement) in the Zeno limit.
-maybe Section 2.2 on the Diophantine equations is more suited to the Appendix.
- The section of "Arrested development" is not clear.
- The Lieb lattice is never defined.
- At the bottom of p.3 the authors compare with Ref.[49] by saying UwaitUdis, but Udis is never defined.
- there are some misprints.

Nevertheless, if the authors think that they can provide much more evidence for their claims, and explain more clearly their results, I could reconsider my decision.

  • validity: low
  • significance: good
  • originality: good
  • clarity: low
  • formatting: good
  • grammar: excellent

Author:  Matthew Wampler  on 2023-01-23  [id 3264]

(in reply to Report 1 on 2022-11-07)

We thank the referee for a close reading of our manuscript and below give a point-by-point response to their remarks.

Referee: This work explores a form of ergodicity breaking, stemming from specific conditions on Floquet dynamics for fermionic systems. The paper focuses on two examples of two-dimensional Floquet systems, where the dynamics are induced by sequentially activating hoppings between the pairs. By looking for the conditions for which the dynamics are frozen, the authors find Diophantine equations which admit some solutions and focus on those regimes. After writing down the conditions for the examples under analysis, the authors discuss the situation in which only some of these conditions are met, which shall result in Hilbert space fragmentation. The paper addresses a timely topic - currently the focus of great attention - and considers two-dimensional models with Floquet dynamics, which have sensible experimental applications. Nevertheless, the paper does not provide sufficient evidence to substantiate its claims and therefore I do not feel to recommend publication in the present form. Let me be more specific. Despite the authors discussing in great detail the examples of the Hubbard-RLBL dynamics and the nearest neighbour's interactions on the Lieb lattice, both from the frozen dynamics and the fragmentation, no numerical evidence is provided on these two examples, but only pictorial representations of the evolution. The only numerical analysis (from which the authors deduce plenty of consequences) is provided for only one system size and leaves a lot of open questions: what would be the result of other local observables? How does the size of the two Krylov subspace scale with N?

Authors: We thank the Referee for their remarks that helped us improve the paper. While the paper is focused on finding exact arithmetic points, additional numerical work is always illuminating. We have added new numerical expositions in several places. These include:

(1) A new section (3.4) investigating entanglement entropy and level statistics in the Hubbard-RLBL in 2D. In the section we show numerical evidence of several different types of subspaces, exhibiting localized dynamics as well as CUE and Poisson level statistics. The numerics is of course limited to a small number of particles, as is done in most papers where such questions are addressed in 2D.

(2) A more “visual” illustration of the exact points for the evolution, where we introduce a new measure to quantifying how close the evolution is to a complex permutation of number states and show how the special points appear in such a plot, Fig (3).

(3) We have added an explanation of the scaling of Frozen states in our 1D hubbard example, section (3.3), showing that the number of frozen states is exponential in system length, however it is at the same time an exponentially small fraction of the total number of states.

Referee: Some other comments - the authors conjecture the existence of solutions for Dmax generic, because a solution is found for Dmax=4. What are the other arguments for this claim?

Authors: We would find it highly surprising if Dmax=4 would be special, given that it already requires the simultaneous solution for the independent primes l=2 and l=3 (thus involving various incommensurate integers in the equations). We added the sentence: “The result for Dmax=4 required simultaneous solution of the equations for two different primes (l=2 and l=3) which suggests the conjecture that there are solutions to the system of equations for any Dmax.”

Referee: - it is not clear what is the difference between Example 2 and Example 3 (studied in Ref. [52] given by nearest neighbour interactions + measurement) in the Zeno limit.

Authors: What we hope to emphasize via example 2 and example 3 is that, despite the fact that the dynamics of the two systems are generally quite distinct (the measurements in example 3 make the evolution non-unitary, for example, in contrast to the completely unitary evolution of example 2), at the special points in parameter space the evolution of the two systems is exactly the same. However, when the system is perturbed away from these points, the dynamics will again begin to differ. For a more in depth discussion of the differences between Floquet and measurement-induced systems such as this, see [Phys. Rev. X 12, 031031]. We have added a remark at the end of section 2.6 to clarify these points.

Referee: -maybe Section 2.2 on the Diophantine equations is more suited to the Appendix.

Authors: Since the topic may not be familiar for some physicists we chose to leave the section in its place, but added the disclaimer that a reader can skip the section to get more directly to the results.

Referee: - The section of "Arrested development" is not clear.

Authors: We have rewritten much of the section on “arrested development”, and hope it is now clearer.

Referee: - The Lieb lattice is never defined. - At the bottom of p.3 the authors compare with Ref.[49] by saying Uwait→Udis, but Udis is never defined. - there are some misprints.

Authors: We have fixed the above and a few other miscellaneous errata in the manuscript.

Referee: Nevertheless, if the authors think that they can provide much more evidence for their claims, and explain more clearly their results, I could reconsider my decision.

Authors: We hope we have clarified the paper sufficiently for the referee to now recommend its publication.

---

## Round 2 · Referee Report · Anonymous (Referee 1) · 2023-2-12

Report

The authors have addressed my remarks.

---

## Round 2 · Author Response

Dear Editor,

We hereby resubmit our paper for your consideration. We thank the referees for a close reading of the manuscript. We believe we have answered the questions of the referees whose inquiries prompted us to improve the presentation, add a new section investigating the entanglement entropy and level statistics of the Hubbard-RLBL model in 2D, introduce a new measure which describes the proximity of a given evolution to a complex permutation of number states, and add a description of the scaling of the number of frozen states in our 1D Hubbard example. In addition, we have also added several smaller clarifying remarks as suggested by the referees. With these additions, and given the positive remarks of the referees regarding the interest and relevance of our results, we believe our paper is now ready to be published in SciPost Physics.

The Authors

---

## Round 2 · List of Changes

• New section (3.4) investigating entanglement entropy and level statistics in the Hubbard-RLBL in 2D
  • In section (2.3) we introduce a new measure which provides a qualitative estimate of the proximity of a given evolution to a complex permutation. Additionally, we provide a plot of this quantity, Fig. 3, in order to better illustrate the appearance of the special Diophantine points in the Hubbard-RLBL model
  • In section (3.3) we add a description of the scaling of the number of frozen states for our 1D Hubbard example
  • We have rewritten much of section (3.1) to improve clarity as requested by referee 2
  • An addition to the summary and discussion section (4) on the effects of long range interactions
  • A remark at the end of appendix A.2 on why m0 (in equation 34) is always even
  • A brief summary (added to section 3.2) of the arguments from Refs 49 and 50 regarding the expectation that disorder might stabilize the dynamics at the special points in some cases
  • A remark (added to section 2.5) on our conjecture that there exist a solution to equation (32) for any Dmax
  • A comment in section 2.6 to clarify the distinction between the NN-RLBL model and the measurement-induced chirality model with nearest neighbor interactions
  • A comment added at the start of section 2.2 suggesting a reader familiar with Diophantine equations may skip the section
  • We fixed a few miscellaneous errata/typos including all of those noted by the referees

---

## Editorial Decision

published